

# Size distribution and source of black carbon aerosol in
# urban Beijing during winter haze episodes
Yunfei Wu[1,*], Xiaojia Wang[1], Jun Tao[2], Rujin Huang[3], Ping Tian[4], Junji Cao[3], Leiming
Zhang[5], Kin-Fai Ho[6], Renjian Zhang[1,*]
Key Laboratory of Regional Climate-Environment for Temperate East Asia,
Institute of Atmospheric Physics, Chinese Academy of Sciences, Beijing, China
South China Institute of Environmental Sciences, Ministry of Environmental
Protection, Guangzhou, China
Key Laboratory of Aerosol Chemistry and Physics, Institute of Earth Environment,
Chinese Academy of Sciences, Xi'an, China
Beijing Weather Modification Office, Beijing, China
Air Quality Research Division, Science Technology Branch, Environment Canada,
Toronto, Canada
The Jockey Club School of Public Health and Primary Care, The Chinese
University of Hong Kong, Hong Kong, China
* Correspondence to: Yunfei Wu (wuyf@mail.iap.ac.cn) and Renjian Zhang (zrj@mail.iap.ac.cn)





**Abstract**
Black carbon (BC) plays an important role in the climate and environment due to its
light absorption, which is greatly dependent on its physicochemical properties
including morphology, size and mixing state. The size distribution of the refractory BC
(rBC) in urban Beijing during the late winter in 2014 was revealed by measurements
obtained using a single particle soot photometer (SP2), when the hazes occurred
frequently. By assuming void-free rBC with a density of 1.8 g cm$^{-3}$, the mass of the rBC
showed an approximately lognormal distribution as a function of the volume-equivalent
diameter (*VED*), for which there was a peak diameter of 213 nm. This size distribution
agreed well with those observed in other urban areas of China. Larger *VED* values of
the rBC were observed during polluted periods than on clean days, implying an
alteration in the rBC sources, as the mass-size of the rBC from a certain source varied
little once it was emitted into the atmosphere. The potential source contribution
functions showed that air masses from the south to east of the observation site brought
a higher rBC loading with more thick coatings and larger core sizes. The mean *VED* of
the rBC presented a significant linear correlation with the number fraction of thickly
coated rBC; the *VED* of the entirely externally mixed rBC was inferred as the *y*-
intercept of the linear regression. This *VED*, with a value of ~150 nm, was considered
as the typical mean *VED* of the rBC from local traffic sources in this study. Accordingly,
the contribution of the local traffic to the rBC was estimated based on reasonable
assumptions. Local traffic contributed 35 to 100% of the hourly rBC mass concentration
with a mean of 59%, during this campaign. A lower local traffic contribution was
observed during polluted periods, suggesting increasing contributions of other sources
(e.g., coal combustion/biomass burning) to the rBC. The heavy pollution in Beijing was
greatly influenced by other sources in addition to the local traffic.
**Keywords**: black carbon aerosol, size distribution, source, haze





## 1 Introduction


Black carbon (BC), the major light-absorbing component in atmospheric aerosols, plays
an important role in the radiative balance of the earth system by directly heating the
lower atmosphere and affecting the cloud cover through semi-indirect effects
(Ramanathan and Carmichael, 2008). Although BC is hydrophobic, it can also act as a
cloud condensation nucleus when internally mixed with other hydrophilic components
(Zhang et al., 2008), indirectly affecting the radiative budget (Ramanathan et al., 2001).
As a result, BC aerosols have a great impact on the regional/global climate and weather
(Menon et al., 2002; Ramanathan and Carmichael, 2008; Ding et al., 2013; Liao et al.,
2015; Huang et al., 2016). Recent research has also illustrated that BC increases
atmospheric stability by its heating effect in the lower troposphere and cooling role at
the surface (Wang et al., 2013). It suppresses the diffusion of pollutants, which
deteriorates the air quality and plays an enhanced role in severe haze (Ding et al., 2016).
However, it is difficult to accurately quantify the radiative forcing and environmental
effects induced by BC because of the high variations in its concentration and
physicochemical properties (IPCC, 2013). The light absorption of BC highly depends
on its size and morphology. Mie calculations for hypothetical BC spheres show that the
mass absorption cross-sections reach their peak at a diameter of ~150 nm and then
decrease sharply with further increases in size (see Fig. 4 in Bond and Bergstrom, 2006).
However, atmospheric BC particles apparently consist of aggregates of small primary
spherules ~15 to 60 nm in diameter (Alexander et al., 2008; Zhang et al., 2008). They
are chain agglomerates when freshly emitted from the combustion sources resulting in
increasing mass normalized absorption with the particle mobility size (Khalizov et al.,
2009). These fresh BC particles are quickly coated by other aerosol components in the
atmosphere, leading to the collapse of the chain agglomerates into more compact BC
cores (Zhang et al., 2008). An alteration in the morphology of BC due to a thin coating
causes competition between light absorption enhancement and decline, resulting in
little variation in the absorption efficiency (Wang et al., 2013; Peng et al., 2016).
Subsequently, the thickened coating of the scattering shell enwrapping the compact BC
cores enhances the light absorption of BC by the lensing effect, although the upper limit





of the enhanced amplitude varied among different studies (e.g., Schnaiter et al., 2005;
Shiraiwa et al., 2010; Khalizov et al., 2009; Peng et al., 2016).
With the rapid development of its economy, China is suffering from heavy air pollution
(Yin et al., 2016). As one of the major aerosol components, the annual BC emissions to
the atmosphere are very high in China, representing approximately half of the emissions
in Asia and one-fifth of the global BC emissions (Qin and Xie, 2012). The mass
concentrations of BC have been widely measured (e.g., Cao et al., 2007; Zhang et al.,
2008), but there is a lack of a comprehensive investigation of the physicochemical
properties of ambient BC aerosols (e.g., size, morphology, and mixing state), due to the
limitations of the measurement methodology. A traditional approach through analyzing
the BC mass of size-segregated aerosol samples has usually been employed to
determine the BC size distribution (Huang and Yu, 2008; Yu et al., 2010). However, it
provides size information on the BC-containing particles rather than on the BC itself
because numerous BC particles are internally mixed with other aerosol components in
the ambient atmosphere (Shiraiwa et al., 2007; Schwarz et al., 2008). Additionally, the
time resolutions of the determined BC size on the basis of this method are typically
hours to days. In the last ten years, a novel analyzer–single particle soot photometer
(SP2)–has provided an advantage to investigate in a highly time resolved manner of the
mass and size of the refractory BC (rBC) (Stephens et al., 2003; Schwarz et al., 2006).
The mixing state of rBC particles can also be derived from the measurement of SP2
(Gao et al., 2007; Moteki and Kondo, 2007, 2008; Laborde et al., 2012). Research on
the sizes and mixing states of rBC based on this technology has been limited to a few
regions in China (e.g., Huang et al., 2012; Wang et al., 2014a, 2015a; Wu et al., 2016;
Gong et al., 2016), as the SP2 is very expensive and its performance is limited (Gysel
et al., 2012; Liggio et al., 2012). It should be noted that the sizes of rBC reported by
SP2 are generally mass-equivalent diameters rather than mobility- or aerodynamic-
based ones, which are determined on the basis of the mass measurements of individual
rBC-containing particles. Thus, they are independent of the morphology or mixing.
Although physicochemical properties of BC in the atmosphere are greatly diverse, its
mass-equivalent sizes should vary little during their typical lifetime in the atmosphere



(~1 week) since BC itself is chemically inert under ambient conditions. In other words,
the mass-size of a BC particle is independent of its morphology and mixing state,
although coating with other components will reduce its mobility diameter and enlarge
the size of the mixed particle in which the BC is embedded. As it is a byproduct of the
incomplete combustion of fossil fuels and biomass, the BC size should be highly
dependent on the emission sources, including fuel type and combustion condition.
Based on the measurement of SP2, Liu et al. (2014) showed smaller sizes of the rBC
cores from traffic than those from solid fuel sources and attributed the rBC
concentrations from the two dominant sources accordingly. The rBC sizes measured at
rural or remote sites were considerably larger than those measured at urban sites (Huang
et al., 2012; Schwarz et al., 2013), implying that smaller sizes of rBC are emitted from
traffic sources. Combining the measurement of SP2 and the chemical source
apportionment of daily $PM_{2.5}$ samples, Wang et al. (2016) showed that the rBC from
biomass burning and coal combustion had larger mass-equivalent diameter than that
from traffic.
Jointly influenced by the local emissions (e.g., traffic exhaust) and regional transport
of air pollutants from the surrounding heavily polluted areas where intense industrial
emissions and coal combustions were reported, the source apportionments of $PM_{2.5}$ and
its subcomponents (e.g., BC) in urban Beijing are highly controversial (Tao et al., 2016;
Zíková et al., 2016). In this study, a novel approach was employed to evaluate the
contribution of local traffic to the rBC concentration in urban Beijing during a
wintertime in 2014 when hazes occurred frequently, on the basis of measurements of
SP2 and reasonable assumptions. Before that, the mass-equivalent size distribution of
rBC in urban Beijing was revealed. The variation in the rBC size was also investigated,
accompanied by an analysis of their chemical compositions and potential source
contributions.

**2 Methodology**
In situ measurements of rBC were conducted using a SP2 (Droplet Measurement
Technology, Inc., Boulder, CO, USA) on the rooftop (approximately 8 m above ground





level) of an experimental building at the Tower Division of the Institute of Atmospheric
Physics, Chinese Academy of Sciences (IAP, CAS), during a late winter period from
24 February to 15 March 2014, before the residential heating was stopped. The SP2
directly detects the incandescent intensity of an individual rBC-containing particle
when it passes through an intra-cavity Nd:YAG laser beam with a Gaussian distribution
(Schwarz et al., 2006). The incandescent intensity is converted to the mass of rBC based
on the calibration of incandescent signals of size-selected soot standards performed
pre/post-sampling. In this study, the Aquadag (Acheson, Inc., USA) was used as a
reference rBC and size-selected by a scanning mobility particle sizer spectrometer
(SMPS; TSI, Inc., Shoreview, MN, USA) for calibration. Compared to the ambient rBC,
it is more sensitive to the incandescence signal. Thus, a scaling factor of 0.75 is
employed with the calibration curve to induce more reliable rBC mass determinations
(Baumgardner et al., 2012; Laborde et al., 2012). Moreover, an approximately 10%
underestimation of the SP2-derived bulk rBC mass concentration due to the detection
limitations outside the rBC mass range of ~0.3–120 fg was considered (Wang et al.,
2014a, 2015a). The total uncertainty in the rBC mass determination was ~25%,
including the uncertainties inherent in the mass calibration, flow measurement and
estimation of BC masses beyond the SP2 detection range (Wu et al., 2016). The
scattering signal is synchronously detected by the SP2 and used to determine the optical
size of a single particle (Gao et al., 2007; Laborde et al., 2012). In this study, the
scattering signal was employed to distinguish the mixing state of rBC-containing
particles. A traditional method based on the delay time between the incandescent and
scattering peaks was utilized to distinguish the rBC cores with and without a thick
coating (Schwarz et al., 2006; Moteki and Kondo, 2007; Wang et al., 2014a; Wu et al.,
2016). On this basis, the number fraction of thickly coated rBC ($NF_{coated}$), defined as
the ratio of the number of thickly-coated rBC particles to that of all detectable rBC
particles, was calculated to characterize the relative mixing extent of the BC aerosols
in different ambient samples. A similar measurement was conducted in January 2013,
and more details of the experimental setup and data process can be found in Wu et al.

166 (2016).

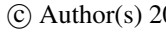

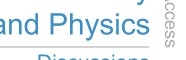
Samples of PM$_{2.5}$ was collected twice a day during this campaign, with each lasting for
twelve hours. The chemical contents including organic carbon (OC), elemental carbon
(EC), water-soluble ions (e.g., SO$_4^{2-}$, NO$_3^-$, and NH$_4^+$) and trace elements were
analyzed in the laboratory, as presented in detail by Lin et al. (2016).

**3 Results and Discussion**
**3.1 Size distribution of rBC and its variation**
As shown in Fig. 1, the mass of rBC (d$M$/dlog$D_p$) exhibits an approximately lognormal
distribution as a function of the volume-equivalent diameter (*VED*) of void-free rBC,
as has been commonly observed (e.g., Schwarz et al., 2006; Huang et al., 2012; Wang
et al., 2016). A minor mode is also captured at large sizes (peaked at ~600 nm), only
accounting for ~6% of the SP2-determined rBC masses. An analogous minor mode was
previously observed at other sites in China. Huang et al. (2011) reported a minor peak
with a diameter of ~690 nm at Kaiping, a rural site in the PRD region of China. Wang
et al. (2014b) found a minor peak with a diameter of ~470–500 nm in a remote area of
the Qinghai–Tibetan Plateau and considered that it was likely a feature of the rBC
distribution of biofuel/open fire burning sources, which needs further measurements
focusing on the size distribution at the emission sources. The peak diameter of the
primary mode, with a value of 213 nm, during the campaign is well within the range
(~150–230 nm) presented by previous studies conducted in different regions (Huang et
al., 2012 and references therein). It should be noted that the density of the assumed
void-free rBC was set to 1.8 g cm$^{-3}$ in calculating the *VED* from the rBC masses
measured in this study, which should result in larger *VED* values compared to those
based on the density of 2.0 g cm$^{-3}$ used in previous studies. If the same density with a
value of 2.0 g cm$^{-3}$ was employed, the peak diameter of the primary mode should be
~206 nm in this study. This value is very close to those observed in urban areas
throughout China, e.g., 210 nm in Shenzhen in South China (Huang et al., 2012), 205
nm in Xi'an in West China (Wang et al., 2015b) and ~200 nm in Shanghai in East China
(Gong et al., 2016). The relatively close mass-size distributions of rBC suggest that
there are similar dominant emission sources in different urban regions in China, where





vehicle exhaust is one of the important sources emitting rBC particles. Compared to
those measured at rural sites in the PRD region in South China (e.g., 220–222 nm,
Huang et al., 2011, 2012), the peak diameters of rBC in urban areas are significantly
lower. This might relate to the greater amounts of coal combustion and biomass burning
around the rural sites (Huang et al., 2012). In contrast, the sizes of the rBC were much
smaller in remote regions, e.g., with a peak diameter of ~175–188 nm in the Qinghai–
Tibetan Plateau area (Wang et al., 2014b, 2015a). Wang et al. (2015a) attributed this
lower peak diameter value to the source and considered that biomass burning generated
a small rBC with peak *VED* values in the range of ~187–193 nm. Another important
reason for the smaller rBC measured in remote regions, in our opinion, is that more
large rBC particles are deposited during their long-range transport to the observation
site. Further research on the sizes of rBC from different sources is needed.
The mass-size distributions of rBC during a polluted day (25 February) and a clean one
(4 March) are also compared in Fig. 1. The average mass concentrations of rBC ($MC_{rBC}$)
were 7.6 μg m$^{-3}$ and 0.4 μg m$^{-3}$, respectively, on the polluted and clean days. The size
distribution of rBC during the polluted day is similar to that during the entire
observation period, although a larger peak diameter was observed, with a value of 221
nm. In contrast, the peak diameter on the clean day is much smaller, with a value of 199
nm. The secondary mode cannot be well characterized on the clean day. As mentioned
above, the mass-sizes of rBC emitted from a certain source change little during their
lifetime in the atmosphere. Thus, the considerable discrepancy of the rBC sizes
illustrates significant source alteration during the polluted period compared to that on a
clean day. Sun et al. (2014) used the measurements of ACSM at an urban site in Beijing
to show that the regional contribution to the BC exceeded 50% during heavily polluted
periods in January 2013. Model simulation also revealed that regional transport
contributed an average of 56% to the PM$_{2.5}$ in Beijing in January 2013 when the hazes
occurred frequently, and even higher during polluted periods (Li and Han, 2016).
Accordingly, regional transport might play an important role in the increase in rBC
sizes during polluted periods in urban Beijing. By comparison, traffic emissions should
be the dominant source of rBC on the clean day, contributing to smaller rBC sizes.





The variation in the *VED* of the rBC is further investigated by comparing the mean *VED*
value of rBC ($VED_{rBC}$) with the mass ratios of secondary inorganic components (i.e.,
ammonium sulfate, AS; ammonium nitrite, AN) to EC, a representation of the aerosol
aging degree. Generally, the average $VED_{rBC}$ of each sample shows an increasing trend,
with increasing ratios of AS to EC (AS/EC) and AN to EC (AN/EC) with correlation
coefficients of 0.63 ($p<0.01$) and 0.61 ($p<0.01$), respectively (Fig. 2a and 2b). Higher
AS/EC and AN/EC values were observed in polluted samples, corresponding to a
higher $VED_{rBC}$ during these periods.
It is interesting to note that the $VED_{rBC}$ correlates more closely with AS/EC than AN/EC,
especially under a certain pollution level. For instance, the correlation coefficient
between $VED_{rBC}$ and AS/EC is 0.88 ($p<0.01$) during clean periods with a PM$_{2.5}$ mass
concentration lower than 35 μg m$^{-3}$ (blue dots in Fig. 2), much higher than that between
$VED_{rBC}$ and AN/EC. By contrast, the $NF_{coated}$ varied less with AS/EC during these
periods (Fig. 2c). This means that a higher AS/EC had less effect on the fraction of
thickly coated rBC during these clean periods but was related to larger rBC sizes, which
were highly dependent on the emission sources. In other words, higher AS/EC values
might indicate an increasing contribution of sources other than traffic to rBC, as sulfur
is one of the major trace elements of coal combustion but not of traffic (Zhang et al.,
2013; Wang et al., 2016), corresponding to larger rBC sizes. On the other hand, $NF_{coated}$
is highly related to AN/EC, with a correlation coefficient of 0.81 ($p<0.01$) during the
clean periods (Fig. 2d). Even for the entire samples, the correlation coefficient between
$NF_{coated}$ and AN/EC can be as high as 0.81 ($p<0.01$), much higher than that between
$NF_{coated}$ and AS/EC, with a value of 0.65 ($p<0.01$). This implies that the mixing state of
rBC is more sensitive to AN/EC in urban Beijing, especially during the clean periods.
The secondary formation of AN might play an important role in the coating processes
of rBC but have a smaller effect on the core size of the rBC.

**3.2 Potential source contribution to rBC mass and size**
The potential source contribution function (PSCF) based on hourly resolved 48-h
backward trajectories arriving at the observation site 100 m above ground level were





performed using TrajStat software (Wang et al., 2009). The threshold of the PSCF
analysis was set to the mean value of each variable. A weight function on the gridded
PSCF values was employed on those cells having few trajectory endpoints (Wang et al.,
2006). Generally, the areas east and south of the observation site had the largest number
of potential source regions of high rBC concentrations, with weighted PSCF (WPSCF)
values of $MC_{rBC}$ larger than 0.7 (Fig. 3a). Previous studies showed that Hebei province,
on the southern and eastern borders Beijing, was a major contributor of pollutants to
Beijing, as its industrial activities are intense (Zhang et al., 2013). The high coal
consumption associated with the heavy industrial activity and residential heating in the
cold season should be an important source of high atmospheric rBC loading in these
areas. Similarly, the distribution of the WPSCF values of $VED_{rBC}$ shows that the eastern
and southern regions are also correlated with large $VED_{rBC}$ values (Fig. 3b). This
implies that the pollution sources in these regions, e.g. heavy industrial activity and
residential heating, tend to produce highly concentrated rBC-containing particles with
large rBC core sizes. The source apportionment of rBC aerosols in London based on in
situ SP2 measurements showed that rBC-containing particles from solid fuel sources
(coal combustion and biomass burning) had significantly larger rBC cores than those
from traffic. Thus, the high WPSCF values of $MC_{rBC}$ and $VED_{rBC}$ in the east and south
might highly correlate to anthropogenic coal/biomass combustion in these regions.
The spatial distribution of the WPSCF values of $NF_{coated}$ is shown in Fig. 3c. Associated
with the aging processes that increase the thickly coating states of rBC-containing
particles through heterogeneous reactions, the WPSCF values of $NF_{coated}$ are generally
high in the areas surrounding the observation site. It should be noted that higher WPSCF
values of $NF_{coated}$ (> 0.7) dominate in the east to south. In addition to the transport of
thickly coated BC particles from these regions, aging processes of locally emitted BC
particles (e.g., from traffic sources) under the southerly dominant condition, in which
the relative humidity (RH) is high (Zhang et al., 2015; Zheng et al., 2015), also increase
the fraction of thickly coated rBC (Wu et al., 2016). Although northerly/northwesterly
winds also blow aged rBC-containing particles with thick coatings, the larger amounts
of non-/thinly coated BC particles from local sources during these periods diminished





the WPSCF values of $NF_{coated}$ in the north to west directions. The low RH and strong
winds from these directions are unfavorable to the coating processes of locally emitted
fresh rBC particles.
The $VED_{coated}$, defined as the $VED$ of those thickly coated rBC cores, shows a dispersive
WPSCF distribution (Fig. 3d). Compared to the distribution of $VED_{rBC}$ with high
WPSCF values that dominate in the east to south, high WPSCF values of $VED_{coated}$ are
located in the northern pathway of air masses being transported to the observation site
as well. This implies that the regional transport of air masses brings large rBC, no matter
which direction it comes from. Dominated by the locally emitted small rBC, the
WPSCF values of $VED_{rBC}$ are low in the northern region. It further illuminates that local
sources such as traffic emit small rBC, while regional transport brings large rBC. On
the basis of the large discrepancy in rBC sizes from local traffic against regional
transport, it is possible to extract the contribution of local traffic emissions from the
mixed rBC sources.

**4 Discussion**
**4.1 Relationship between rBC size and mixing state**
As large rBC sizes are usually accompanied by significant contributions of regional
transport, which also lead to a high fraction of thickly coated rBC, the $VED_{rBC}$ is directly
compared with the $NF_{coated}$ as shown in Fig. 4. The two-dimensional histogram of the
5-min average $VED_{rBC}$ and $NF_{coated}$ presents a significant linear correlation between the
two variables. It is characterized more clearly by the variation in the mean $VED_{rBC}$
values averaged in increased $NF_{coated}$ bins with a resolution of 2% (magenta circles in
Fig. 4). The observed minimum value of the 5-min $NF_{coated}$ is ~10%, representing that
there is little completely external mixing of rBC in the ambient atmosphere, even for
short periods. However, an assumed mean $VED$ of completely externally mixed rBC is
extrapolated from the linear curve to $NF_{coated}$ with a value of 0% (i.e., the *y*-intercept
value). This inferred $VED$, with a value of ~150 nm, might be considered as the typical
mean $VED$ of freshly emitted rBC from vehicle exhaust, which is little coated (Zhang
et al., 2008; Peng et al., 2016). We are surprised to find that the linear relationship





between $VED_{rBC}$ and $NF_{coated}$ seems to be common, as indicated by an almost identical
result observed in another campaign conducted in January 2013 (Wu et al., 2016) (gray
circles in Fig. 4). More observations are needed to verify this relationship. However,
according to the results presented in this study, a mean $VED$ of ~150 nm is legitimately
accepted as the typical SP2-determined mean $VED$ of fresh rBC from local traffic
sources. As mentioned above, the $VED$ of certain rBC varies little once it is emitted to
the atmosphere. Thus, the mean $VED$ with a value of ~150 nm was employed in this
study as the representative of the rBC size from local traffic.
The variation in $VED_{coated}$ with $NF_{coated}$ is also shown (magenta triangles in Fig. 4). It is
interested to find that, compared to $VED_{rBC}$, $VED_{coated}$ presents a fluctuant variation as
$NF_{coated}$ increases. The larger $VED_{coated}$ at lower $NF_{coated}$ is comprehensible because
regionally transported large rBC dominates in the thickly coated rBC particles, and the
small rBC from local traffic is mainly externally mixed with other aerosol components
at this stage. As the $NF_{coated}$ increases from 10–20% to 30–40%, the mean $VED_{coated}$
gradually decreases from ~200 nm to ~190 nm. This implies that some small rBC (e.g.,
rBC from local traffic) contributes a considerable part of the thickly coated rBC
particles at this stage. In addition to the influence of the emission sources on the rBC
size, this decrease in $VED_{coated}$ can also be explained by the contamination of the local
traffic emitted small rBC into the thickly coated rBC particles through atmospheric
aging processes (i.e., coating with other components). It should be noted that the
$VED_{rBC}$ sustained increases at this stage, implying that other sources besides the local
traffic also brought large rBC at the same time. This is because if the increase in $NF_{coated}$
only results from the coating processes of the local traffic emitted rBC, the $VED$ of the
entire rBC (i.e., $VED_{rBC}$) should vary little. The $VED_{coated}$ increases significantly when
$NF_{coated}$ exceeds 40%, suggesting that regional transport dominates at this stage,
bringing a large amount of thickly coated rBC particles with a large rBC core.
Meanwhile, the mean $MC_{rBC}$ increases dramatically from 1.3 µg m$^{-3}$ to 5.0 µg m$^{-3}$ when
$NF_{coated}$ increases from 30% to 50%, further confirming the great contribution of
regional transport to the rBC at this stage. By comparison, the mean rBC concentration
varies less in the range of 0.8–1.4 µg m$^{-3}$ when $NF_{coated}$ is lower than 30%. The



observation from the campaign of 2013 shows a similar variation in $VED_{coated}$ against
$NF_{coated}$ (gray triangles in Fig. 4).

**4.2 Extracting the local traffic contribution to rBC**

As $VED_{rBC}$ with a value of ~150 nm is expected to be the typical mean $VED$ of the local
traffic emitted rBC and varies little in the atmosphere, it provides the possibility of
extracting the contribution of the local traffic to the rBC from the total rBC mass
concentration according to the variation in $VED_{rBC}$. However, the typical mean $VED$ of
rBC from other sources, such as coal combustion and biomass burning, is difficult to
identify. It is dependent on many factors including fuel type and combustion condition.
In this study, a simple assumption was employed to identify the typical mean $VED$ of
rBC from other sources besides local traffic according to where the air masses were
from. During a short period when the source emissions are relatively stable, the rBC
from a certain direction was assumed to have a certain mean $VED$, no matter from which
source it is emitted. Thus, a cluster analysis was performed on the 48-h backward
trajectories that arrived at the observation site. Five clusters were identified using
TrajStat software according to the total spatial variation in the cluster numbers (as
shown in Fig. S1). As the rBC tends to be more coated in the regionally transported air
masses, the mean $VED$ of the rBC from sources other than local traffic was derived
from the values of $VED_{coated}$. The local traffic emitted small rBC also can also become
thickly coated through aging processes in the atmosphere, so a further assumption is
employed to consider the $VED$ of rBC from other sources equal to the mean value of
the upper 5% percentile of $VED_{coated}$ in each cluster. Five typical mean $VED$s of rBC
from sources other than local traffic were identified, with values in the range of 195.5–
208.3 nm (Fig. S1). Such a simple assumption might have an impact on the absolute
contribution of the local traffic to the rBC, but it should well reflect the variation in the
traffic contribution.
Accordingly, the hourly-resolved traffic contribution to the rBC was extracted on the
basis of the derived $VED$ of the rBC from local traffic and other sources. The mass
fraction of the traffic-induced rBC ($MF_{traffic}$) is shown in Fig. 5a (red line). During this





campaign, approximately 35% to 100% of the hourly $MC_{rBC}$ is attributed to local traffic
emissions, with a mean of 59%. Based on a multiple linear regression analysis of the
contributions of the three dominant factors (i.e., traffic, coal combustion and biomass
burning) to the rBC derived from the chemical source apportionment of the daily $PM_{2.5}$
samples, Wang et al. (2016) showed a slightly lower contribution of the traffic to the
rBC in urban Xi'an, with a mean of 46% and a daily contribution in the range of 0.8 to
77.2%. Since entirely different methods were employed in addition to the different
locations, the resolved traffic contribution to the rBC should not be compared absolutely.
However, the relatively lower $MC_{rBC}$ in this study (with a mean of 2.8 µg m$^{-3}$ compared
to 8.0 µg m$^{-3}$) might partly interpret the slightly higher contribution of traffic, as a lower
$MC_{rBC}$ is usually accompanied by a higher contribution of the local traffic. It is clear
that $MF_{traffic}$ is negatively correlated with $MC_{rBC}$, with the correlation coefficient as high
as -0.84 (p<0.01) between the daily moving averaged $MF_{traffic}$ and $MC_{rBC}$ (Fig. 5a). This
means that the traffic contribution to the rBC decreased significantly during the polluted
periods when the rBC loading increased. In other words, the rBC from other sources
such as coal combustion and biomass burning play an increased role in these polluted
periods. This implies that the high $MC_{rBC}$ in urban Beijing was not only due to the
accumulation of the local traffic emissions during stable synoptic conditions but also
be attributed to the overlaying pollution from other sources.
The diurnal variations of the decomposed $MC_{rBC}$ from local traffic and other sources
are shown in Fig. 5b and 5c, respectively. A common diurnal variation in $MC_{rBC}$ with
high values during the nighttime and low in the daytime is shown by both the traffic
and other sources producing rBC, suggesting the important impact of the mixing layer
height on the surface $MC_{rBC}$. A high mixing layer in the daytime, especially in the
afternoon, favors the diffusion of the pollutants, leading to a low value of $MC_{rBC}$. A low
mixing layer in the nighttime suppresses the diffusion of pollutants, resulting in a high
value of $MC_{rBC}$. It is noted that a significant peak $MC_{rBC}$ of local traffic was observed
in the early morning (05:00–06:00 local time). Moreover, the increase in the local traffic
related $MC_{rBC}$ occurs earlier than that of other sources in the evening. It corresponds
well to the increased traffic contribution in the morning and evening rush hours. To

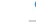

some degree, the diurnal variation verifies the rationality of the method we employed
to distinguish the contribution of the local traffic emission from that of other sources.

**5 Summary and Concluding Remarks**
An approximate lognormal size distribution of the rBC in volume-equivalent diameter
in urban Beijing during a polluted wintertime in 2014 was observed on the basis of
measurements using a SP2. The peak diameter was 213 nm, assuming void-free rBC
with a density of 1.8 g cm$^{-3}$, which is close to the values observed in other urban areas
in China. The measured sizes of the rBC were considerably larger during the polluted
period than in the clean period, implying a source variation of the rBC. The mean
$VED_{rBC}$ was positively correlated with the ratios of secondary inorganic aerosols
(including AS and AN) to EC, more significantly with AS/EC, especially at a certain
pollution level. This implies that the rBC sizes are highly related to the emission sources
because sulfur is one of the major trace elements in coal combustion, while little is
emitted from traffic. By comparison, the mean $NF_{coated}$ was correlated more with
AN/EC, implying the important effect of the secondary formation of nitrate on the rBC
mixing state. The PSCF analysis showed that regional transport from the east to south
of Beijing was a major source of high rBC loading in Beijing and accompanied by a
large $VED_{rBC}$ and high $NF_{coated}$.
The relationship between $VED_{rBC}$ and $NF_{coated}$ was further discussed. A significant
positive correlation existed among the two variables. The mean $VED$ of the entire
externally mixed rBC was extrapolated from the linear curve to $NF_{coated}$ being equal to
0. The inferred $VED$ with a value of 150 nm was considered as the typical mean $VED$
of the rBC from local traffic. Based on the inferred $VED$ and further reasonable
assumptions, the local traffic contribution to the rBC was extracted using a multiple
linear regression to $VED_{rBC}$. Traffic emissions played an important role in the rBC
loading in urban Beijing, contributing 59% of the $MC_{rBC}$, on average, in the campaign.
However, its contribution decreased significantly in the polluted period. A significant
negative correlation is found between the daily moving average $MC_{rBC}$ and $MF_{traffic}$
with a coefficient of -0.87. A similar diurnal variation in the decomposed $MC_{rBC}$



associated with local traffic and other sources was observed with high values in the
nighttime and low in the daytime. However, a significant increase in traffic $MC_{rBC}$ was
observed in the early morning and evening, indicating the increased contribution of
local traffic emissions. Although the absolute contribution of the local traffic might be
not entirely accurate in this study, as inferences and assumptions are employed, its
relative variation is still clear. Further research on the size measurement of rBC directly
from varied sources, including coal combustion, biomass burning and traffic exhaust,
is needed to validate our work. This work provides a relatively simple but novel method
to extract the contribution of the local traffic to the rBC on the basis of the size
measurement of the rBC in an ambient atmosphere. This work should be meaningful to
source apportionment research in urban Beijing where the air pollution is quite severe.

**Acknowledgments:**
This work was supported by the National Natural Science Foundation of China (No.
41575150, 41305128), the Special Scientific Research Funds for Environment
Protection Commonweal Section (No. 201409027) and the Jiangsu Collaborative
Innovation Center for Climate Change.





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





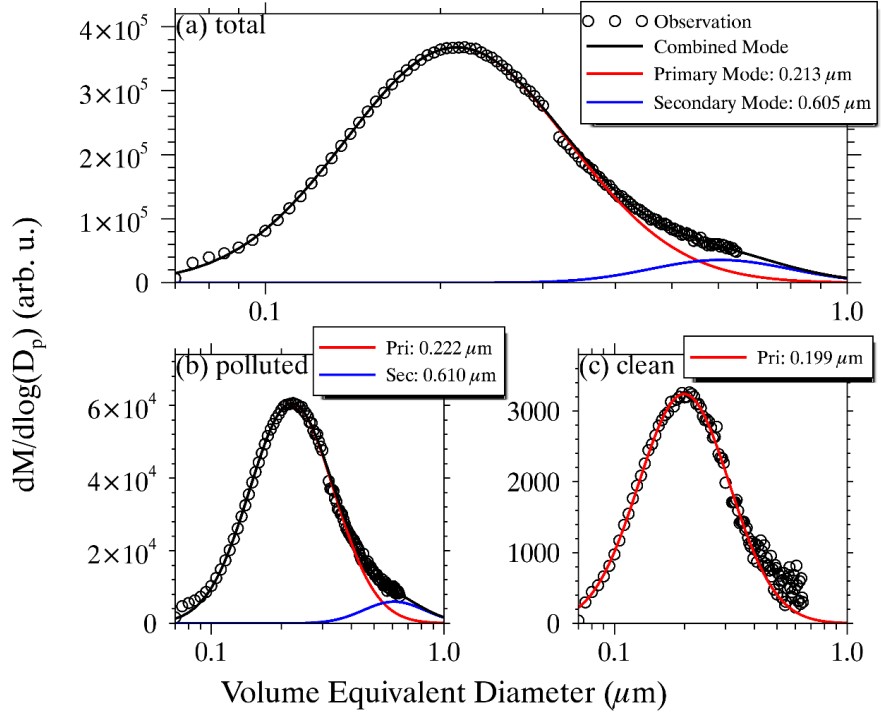

Fig. 1. Size distributions of rBC in volume-equivalent diameter during a campaign from 24 February

to 15 March, 2014. The red and blue lines are the lognormal fittings to the primary and secondary

modes, respectively, and the black ones correspond to the combined mode.





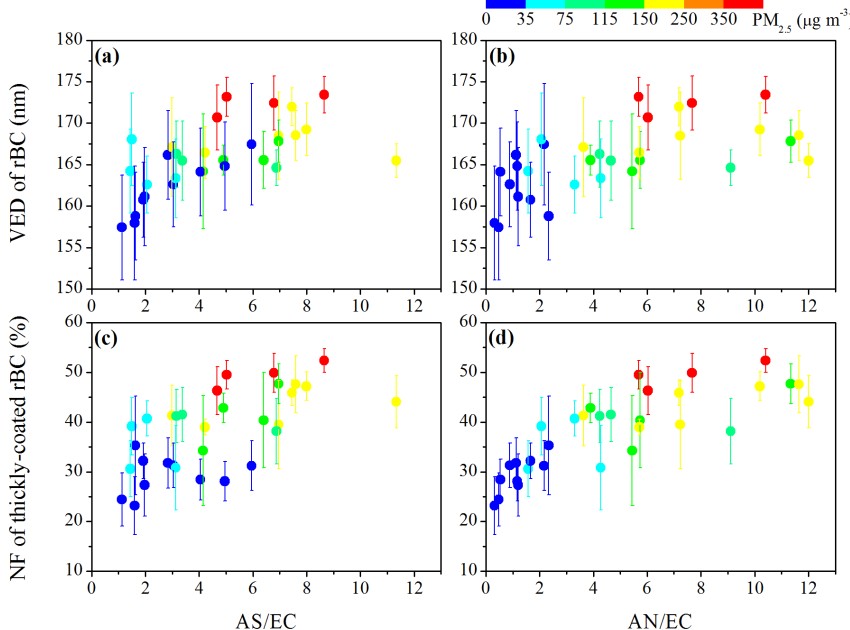


Fig. 2. Variation in the average volume-equivalent diameters of rBC ($VED_{rBC}$) as a function of the
mass ratios of (a) ammonium sulfate (AS) and (b) ammonium nitrite (AN) to elemental carbon (EC).
The same apply for (c) and (d), but for the number fraction of thickly coated rBC ($NF_{coated}$). The
vertical bar denotes one standard deviation. The color scale represents the pollution levels defined
as the PM$_{2.5}$ mass concentration according to the AQI standard of MEP of China.





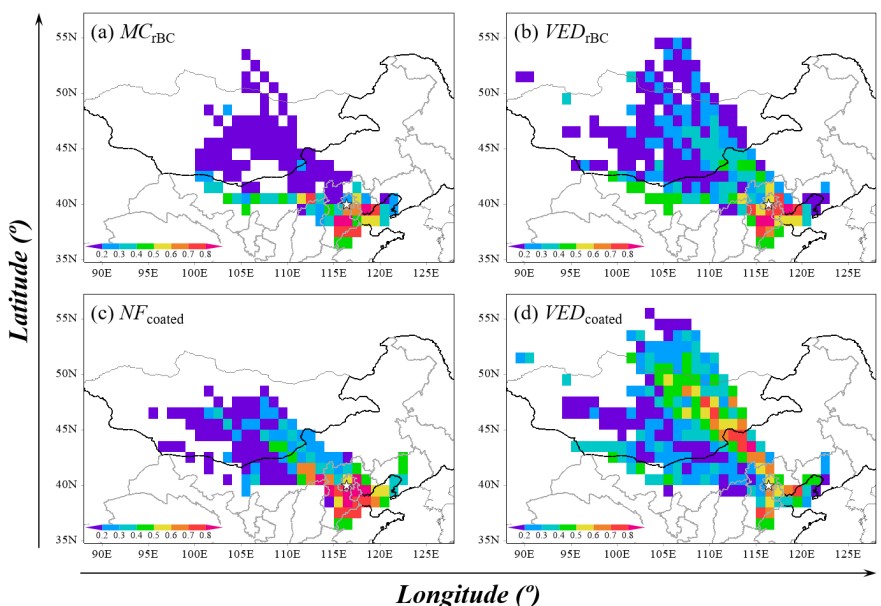


Fig. 3. Distributions of gridded (1°×1°) potential source contribution functions of (a) mass

concentration (*MC*) and (b) volume equivalent diameter (*VED*) of rBC, and (c) number fraction (*NF*)

and (d) *VED* of thickly coated rBC. The overlaid star symbol represents the geographical location

of the observation site.

646





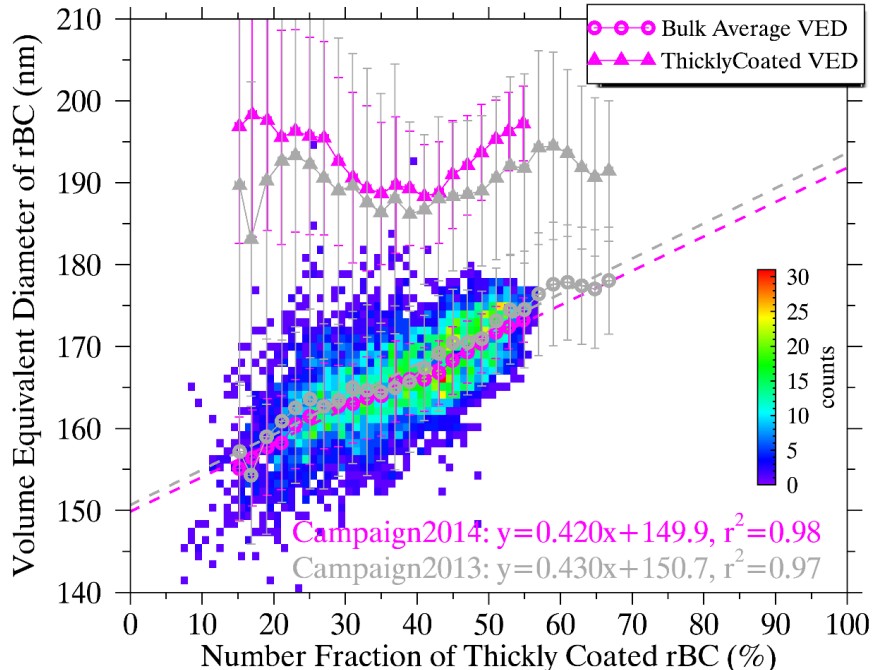

Fig. 4. Two-dimensional histogram of the 5-min average volume equivalent diameter of rBC

($VED_{rBC}$) against number fraction of thickly coated rBC ($NF_{coated}$) during the campaign in the late

winter in 2014. The magenta circles and triangles with error bars represent the mean $VED_{rBC}$ and

$VED$ of thickly-coated rBC ($VED_{coated}$) averaged in each $NF_{coated}$ bin with a resolution of 2%,

respectively. The dashed magenta line denotes the linear regression of $VED_{rBC}$ against $NF_{coated}$. The

relationship between $VED_{rBC}$ and $NF_{coated}$ during another campaign in January 2013 (Wu et al., 2016)

is comparatively overlapped in gray.





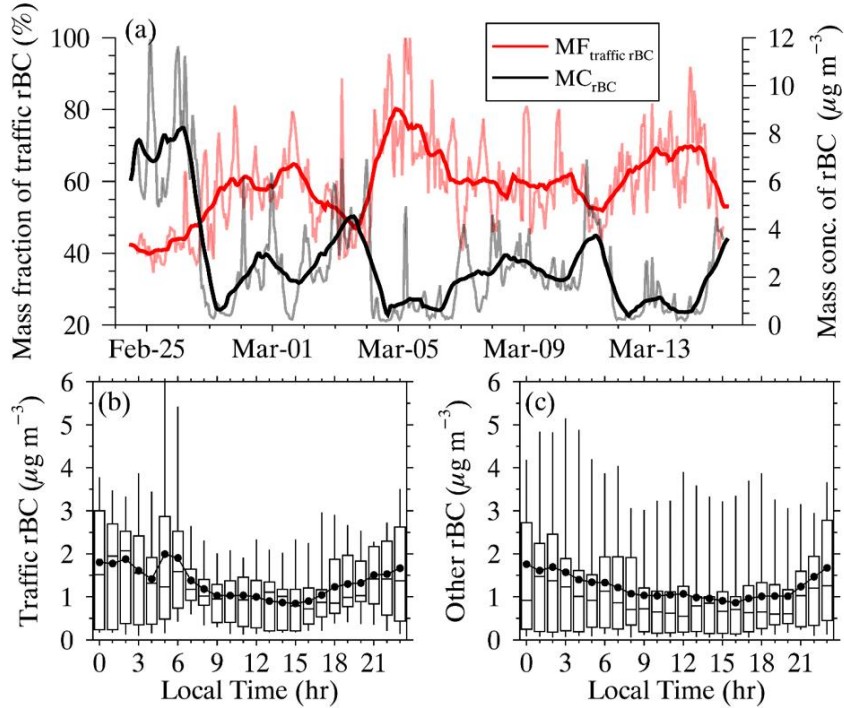

656

Fig. 5. (a) Time series of hourly mass concentration of rBC ($MC_{rBC}$) and mass fraction of local traffic

related rBC ($MF_{traffic}$). The bold lines represent the variations of the daily moving averaged $MC_{rBC}$

and $MF_{traffic}$. (b) and (c) show the diurnal variations in the decomposed rBC from local traffic

emission and other sources, respectively.