# Peer review of "Size distribution and source of black carbon aerosol in"

_Atmospheric Chemistry and Physics, 2016_

## Referee Comment (RC1) · Anonymous Referee #1 · 12 Feb 2017

The paper by Wu et al. is focused around BC bearing particle properties measured by state of the art instrument SP2. The authors did a reasonable job in trying to apportion BC to traffic and coal/biomass combustion sources, however, due to number of assumptions used the result is quite uncertain and unfortunately no validation is available. The paper can be considered for publication after addressing the comments below. Overall, the paper is fairly well written and executed, although a better job is expected regarding the uncertainties.

Major comments

It looks like the whole source apportionment is centered on the regression analysis of VED(rBC) versus thickly coated particles, resulting in traffic related rBC particles of 150nm in size. This result is very central to the main findings of the study – source

contribution of traffic related BC particles. However, no justification whatsoever is provided what defines thickly coated particle and, consequently, what impact it would have if criterion of thickly coated particle is varied (I believe the criterion is the ratio of equivalent diameters of the core and the particle). Overall, the method based on regression analysis is neglecting the fact that particles are rarely externally mixed, except very close to the source which in case of traffic is a car tailpipe. Further away from the source, spatially and temporarily, particles become internally mixed and sources combined (through coagulation, secondary processes and deposition) making it very difficult to justify whether 150 nm particle is indeed traffic related or some of them advected from a population of coal combustion particles (sub-population of smaller coal combustion particles). There is no firm justification that 150 nm particles are indeed originating from traffic only (why not e.g. 120 nm) and indirect evidence provided is not sufficient.

Second major problem is the absence of method validation. One obvious validation would be radiocarbon analysis if that was considered at the start of the study. As it stands, the authors should at least do a thorough analysis of the studies (primarily but not exclusively in China) where radiocarbon analysis has been done and BC has been isotopically apportioned. If the SP2 measurements were combined with radiocarbon analysis it would constitute a significant advancement.

Minor comments

Line 32. Why exactly VED would vary little once rBC is emitted into atmosphere when the time scale of secondary processes is many hours or as long as the particle is airborne and particles undergo cloud processing and dry/wet deposition?

Line 129. Introduce assumption(s) briefly here.

Line 162. Provide a number for defining thickly-coated rBC particles. What constitutes "thickly"?

Line 167. "were" instead fo "was".

Line 180. When something mentioned for the first time, spell it out. PRD = Pearl River Delta?

Line 188. What was the reason of choosing particular density?

Line 195. The relatively similar....

Line 207. This is clearly the most likely reason for different VED values. Consider also cloud processing (∼15min time scale) initiating wet deposition.

Line 214. It is not much smaller, only smaller by 10%.

Line 230. Generally, VED(rBC) were positively correlating with AS/EC and AN/EC ratios....

Line 242. Availability of ammonia is most likely responsible for such pattern, because sulphuric acid is neutralized first and only then nitric acid (acid strength effect) if there is enough ammonia. Neutralisation with ammonia is a passive process. Try looking at differences in the degree of neutralisation.

Line 252. How could secondary AN have any effect on the core size of rBC when formation processes of the two are completely different?

Line 274. Reference London study.

Line 315. "which has little coating..."

Line 316. Why is this surprising as large VED(rBC) would be formed in the presence of copious amounts of gaseous precursors contributing to thick coating during atmospheric processing (secondary formation)?

Line 351. The method ignores the fact that local traffic VED is including contribution of other sources unless the authors have access to specific experiments proving the traffic VED. Considering contribution of other sources to VED makes local traffic contribution biased high. The method works for the estimating the upper limit, but the real contribution can be very different without proper measurement of traffic emitted VED(rBC).

Line 357. This is a very bold assumption when BC sources are not localized, but instead present in every region. Differences in VED in different regions depend on relative contribution of sources, but never fixed. Also consider heterogeneity of traffic sources as car fleet in major cities is very different from rural ones.

Line 370. Five typical VEDs were identified in VED range of 5% only? This is unreasonable.

Line 435. Those values must correlate, because they are methodologically related.

Fig.1. Why there is a gap in the spectrum?

Fig.5 (and respective text). Sources are poorly separated, because of high traffic contribution around midnight. That is unreasonable and points at overestimated traffic contribution.

[Figure]

---

## Referee Comment (RC2) · Anonymous Referee #2 · 17 Mar 2017

Article represents results from a short (3 week) measurement campaign using a SP2 black carbon analyzer and a small amount of filter samples. Measurements are done on roof top of the institute at central Beijing. Based on these results and trajectory analysis, authors have developed a method for estimating the contribution of black carbon originating from a local traffic. Article is well written and easy to understand. Unfortunately the scientific novelty value of these results is poor. A very similar dataset was recently published by Wang et al., 2016, Atmospheric Environment. Article by Wang et al includes also three weeks of SP2 measurements, done right before this campaign in Beijing, partly by the same institutes. Also, several other articles containing SP2 measurement results have been published from Beijing and urban areas in Asia with longer data time series and much larger set of instruments. These already include the main results of this article. The developed method for analyzing the contribution of

local traffic is poorly described and verified. Based on this article, it is hard to estimate if it works.

In order to publish this authors would need to significantly improve this article and include results with scientific novelty value. Including e.g. a longer dataset with data collected during different seasons and a larger set of instruments would definitely help. Also, authors should prove that the developed method works by comparing results of this method to other independent measurement data. Also, uncertainty estimate and overall validation of the method would be necessary. Also, would be important to state if authors think that this developed method will be applicable to other locations or only to Beijing. A literature review/comparison of these results to previously published BC size distributions and existing source analysis methods would also be needed. Also would be useful if authors could explain why the diurnal cycles of the BC from both traffic and other sources are highest during nighttime? I would assume that at least for local sources, maxima would be during morning or evening rush-hour? Also, uncertainty estimation for BC results and especially for the larger mode would be important addition. The large BC mode (with maximum at around 500nm) is very small and very close to the upper particle size limit of the instrument, so it would be important to estimate how real this BC mode is. Lastly, the title and aim of this article are slightly conflicting. Title suggest that this article focuses on BC size distributions and sources, whereas the aim states that the aim was to develop a novel approach to evaluate the contribution of local traffic to the rBC concentration. Might be useful to include the method development to the title, if that is the main goal.

---

## Author Comment (AC1) · 27 Apr 2017

We greatly appreciate the reviewer for providing very constructive comments which have helped us improve the paper. We have considered the comments carefully and revised the manuscript accordingly, as detailed below in our point-to-point responses to the specific comments.

Major comments 1. It looks like the whole source apportionment is centered on the regression analysis of VED (rBC) versus thickly coated particles, resulting in traffic related rBC particles of 150nm in size. This result is very central to the main findings of the study – source contribution of traffic related BC particles. However, no justification whatsoever is provided what defines thickly coated particle and, consequently, what impact it would have if criterion of thickly coated particle is varied (I believe the criterion

is the ratio of equivalent diameters of the core and the particle). Overall, the method based on regression analysis is neglecting the fact that particles are rarely externally mixed, except very close to the source which in case of traffic is a car tailpipe. Further away from the source, spatially and temporarily, particles become internally mixed and sources combined (through coagulation, secondary processes and deposition) making it very difficult to justify whether 150 nm particle is indeed traffic related or some of them advected from a population of coal combustion particles (sub-population of smaller coal combustion particles). There is no firm justification that 150 nm particles are indeed originating from traffic only (why not e.g. 120 nm) and indirect evidence provided is not sufficient.

Response:

(1) The reviewer is concerned with the definition of 'thickly coated rBC-containing particles', and suspects the definition might have impact on the results presented in our manuscript. In our initial manuscript, we simply introduced the method to define the 'thickly coated rBC-containing particles' according to the delay time between SP2 detected incandescent and scattering signal peaks of individual particle. Since this method was described in detail in literature (Schwarz et al., 2006; Moteki and Kondo, 2007) and has been widely used in SP2 related studies (e.g., Wang et al., 2014; Wu et al., 2016; ...), there is no need for a detailed description in our study. We have added a short description in the revised manuscript to make the definition of 'thickly coated rBC-containing particles' clearer, which reads: "The rBC-containing particles were defined as either thickly coated or uncoated/thinly coated according to the distribution of detected lag times, which was bimodal and had a local minimum at 2 $\mu$s (Fig. S1 in the supplemental files). We defined the rBC particles as thickly coated if the lag times were longer than 2 $\mu$s" (Lines 162–166 in the revision). In addition, the method to distinguish the mixing status of rBC-containing particles used in this study is traditional and qualitative, which don't need the assumptions on the particle's physical and chemical properties (e.g., morphology and refractive indices). This method cannot give

the exact diameter of the particles, which is different from the method according to the equivalent diameter of rBC core and optical size of the entire particle. The delay time method can only estimate whether the particle is thickly coated and cannot provide how thick the coating is. Although the definition of 'thickly coated' based on the delay time method is related to the ratio of equivalent diameters of the core and the particle, it is not entirely decided by the absolute ratio of the diameters. Moteki and Kondo (2007) indicated that at least a coating thickness of 70% was needed to reach the delay time threshold for 'thickly coated particles'. Laborde et al. (2012) also illustrated the delay time threshold was expected to occur somewhere in the range of ∼70% coating volume fraction. The reviewer mentioned that particles are rarely externally mixed, except very close to the source which in case of traffic is a car tailpipe. We also illustrated that the rBC particles freshly emitted from traffic can be quickly coated (Line 70 in our initial manuscript). It took ∼4.6 hours for the particles thickly coated by other component under the ambient conditions in Beijing (Peng et al., 2016). In this study, we are concerned about the sizes of rBC which are relatively stable in the atmosphere. The mixing states of rBC particles provide an auxiliary description in the discussion on the sources of rBC. We don't focus on the mixing state of rBC particles and how thick the coating is. Thus, the qualitative method on the basis of delay time between incandescent and scattering signal peaks was employed. (2) As the reviewer proposed, there is indeed no further firm justification to prove the typical mass equivalent diameter of rBC is ∼150 nm for traffic source at current stage of our study. We tried to investigate the sizes of rBC directly measured using the similar method at typical sources, such as traffic, coal/biomass combustion from existing literatures. Unfortunately, to the best of our knowledge, we cannot find available supporting materials. However, the mass size distribution of elemental carbon (EC) from the size-segregated sample revealed a mode with peak at 150 nm in aerodynamic diameter for freshly emitted EC (Yu and Yu, 2009; Yu et al., 2010) in PRD region. Huang et al. (2006) reported a relatively smaller accumulate EC mode with peak at ∼120 nm. However, in these studies, a much larger traffic-related EC size was also observed with peak diameter at ∼400 nm,

which was seldom observed in the urban area and tunnel in developed countries. Yu et al. (2010) considered this characteristic of EC size distributions is related to the result of high engine loads and low combustion efficiencies in Chinese vehicles. We don't know whether the size distribution of EC in Beijing is similar to that in PRD region as there were few studies measured the EC in such small sizes (i.e., with diameter < ∼400 nm). Considering that the urban location of our site and the stringent fuel and vehicle emission standards implemented in urban Beijing, relatively higher combustion efficiencies can be expected, resulting in a smaller EC mode. On this basis, we considered the smaller EC mode with diameter of ∼150 nm should be dominated at our site, although further direct measurement is needed. In the revised manuscript, we have added the citation of these relevant literatures (Lines 334–345 in the revision). In our study, this typical local traffic-related VED with diameter of ∼150 nm was extrapolated from the linear regression of VEDrBC against NFcoated. The physical meaning can be expected from the extrapolation as the freshly traffic-emitted rBC are rarely thickly coated. During the analysis of experimental data, we also suspected the robustness of this linear regression and the deductive VED of traffic-emitted rBC. Thus, we analyzed the data obtained in another independent campaign conducted in winter 2013. It was very interesting to find there was a similar linear relationship between VEDrBC and NFcoated. Meanwhile, the slope and y-intercept values of the linear regression in the two independent campaigns were almost the same (Fig. 4). On this basis, we considered that the similar linear regression might be commonly existed, although more observations are needed to further confirm this result. Meanwhile, the direct measurement of rBC sizes at the traffic emission source is also needed to give a more reliable evidence. At the current knowledge status, the assumption made in this study should be acceptable.

2. Second major problem is the absence of method validation. One obvious validation would be radiocarbon analysis if that was considered at the start of the study. As it stands, the authors should at least do a thorough analysis of the studies (primarily but not exclusively in China) where radiocarbon analysis has been done and BC has been
isotopically apportioned. If the SP2 measurements were combined with radiocarbon analysis it would constitute a significant advancement.

Response: This comment proposed by the reviewer is meaningful and farsighted. It provides a direction for our further studies on BC sizes and apportionment. Unfortunately, we didn't perform the radiocarbon analysis in this study. Although the aerosol samples were collected in the campaign, there were not enough carbon amount for radiocarbon analysis in the laboratory as a low-volume sampler operated at a nominal flow rate of 16.7 L/min was utilized. According to the reviewer's suggestion, we have reviewed literature associated with BC apportionment based on radiocarbon analysis. The 14C analysis was usually used to distinguish the fossil-fuel combustion and biomass burning elemental carbon (EC). Zhang et al. (2015) revealed 76% EC was attributed to fossil-fuel combustion in urban Beijing in the extreme winter haze episode of 2013 on the basis of 14C analysis. The 14C analysis for the samplers collected at a suburban site of Beijing before/during/after the Asia-Pacific Economic Cooperation (APEC) summit held in November 2014 revealed a relatively higher contribution of biomass burning to EC (36% / 46% / 33%). The significant decrease in the contribution of fossil-fuel combustion during APEC was related to the control of industry and traffic emissions (Liu et al., 2016). However, it is difficult to distinguish the traffic-related source from the fossil-fuel combustion based on the 14C analysis. Stable carbon isotopes of traffic source showed a distinguished range from coal combustion (Widory, 2006; Cao et al., 2011), it was also very difficult to identify the exact contribution of traffic to EC due to the wide range of stable carbon isotopes. Generally, the radiocarbon analysis is advantage to distinguish the fossil-fuel and biomass burning source, but incapable of separating traffic-related sources from the fossil-fuel combustion. On the basis of SP2 measurement, the contribution of fossil-fuel combustion (traffic and coal burning) contributed ∼80% of rBC in Xi'an (Wang et al., 2016), close to the apportionment result based on the 14C analysis (78%, Zhang et al., 2015). The apportionment results were comparable, even the completely different approaches were employed. The traffic related rBC was determined in Wang et al. (2016), with a mean contribution

of 46%. It was not much different from our result where the local traffic contributed 59% to rBC on average. We will try to combine the two methods (i.e., SP2 and radiocarbon analysis) in future studies to give a more specific and reliable apportionment of BC.

Minor comments 1. Line 32. Why exactly VED would vary little once rBC is emitted into atmosphere when the time scale of secondary processes is many hours or as long as the particle is airborne and particles undergo cloud processing and dry/wet deposition?

Response: We meant that the VED of an individual rBC particle varied little during its lifetime in the atmosphere because the rBC is chemically inert under ambient condition. Although cloud processing will change the physical/optical size of the rBC-containing particle through coating of other components (e.g., organic matters, sulfate, . . .), the mass-size (i.e., the VED) of the rBC core varies little. Moreover, the typical lifetime of BC is ∼1 week in the atmosphere (Bond et al., 2013), and ultimately removed from the atmosphere through wet (e.g., in precipitation) and dry deposition to the Earth's surface. Because there is no industrial rBC emission in urban Beijing, traffic should be one of the most important local sources of rBC. The emission intensity of vehicles is relative stable, resulting a constant VED of rBC measured in our study if other sources effect little. Thus, the variation in VED of rBC should be greatly interpreted by the alteration in rBC sources.

2. Line 129. Introduce assumption(s) briefly here.

Response: We have illustrated the assumptions in our revised manuscript according the comment, which reads: "a novel approach was employed to evaluate the contribution of local traffic to the rBC concentration based on the measured rBC sizes and reasonable assumptions including a deductive mean diameter of rBC from local traffic and relatively stable rBC sizes in the air masses transported over certain regions." (Lines 129–133 in the revision)

3. Line 162. Provide a number for defining thickly-coated rBC particles. What constitutes "thickly"?

[Figure]

Response: As we mention in the response to the first major comment, the 'thickly coated rBC particle' defined here was a qualitative conception. It was distinguished using the delay time between SP2 detected incandescent and scattering signal peaks of an individual particle, instead of directly characterize the size of rBC and its coating. This method is relatively easy to operate and doesn't rely on the assumptions of particle physical/chemical properties, which has been widely used in SP2-related studies (e.g., Moteki and Kondo, 2007; Wang et al., 2016). Previous studies revealed that at least a coating thickness of 70% was needed to reach the delay time threshold for 'thickly coated particles' (Moteki and Kondo, 2007; Laborde et al., 2012). As the observation site is not close to the emission sources (e.g., traffic road), naked rBC without coating should be seldom observed. Due to the quick coating processes of freshly emitted rBC from traffic, there should be a great part of thinly coated rBC under traffic dominant condition. These thinly coated rBC particles were classified together with the naked ones, significantly distinguished from the thickly coated ones which were underwent sufficient coating processes in the atmosphere or from sources such as coal combustion/biomass burning. Once the regional transport dominated, a great fraction of thickly coated rBC should be observed because the rBC particles were initially thickly coated from coal combustion/biomass burning sources and suffered from coating processes during their transport to the observation site. We don't concern about how thick the coating is in this study.

4. Line 167. "were" instead of "was".

Response: Corrected.

5. Line 180. When something mentioned for the first time, spell it out. PRD = Pearl River Delta?

Response: Corrected. We have also checked throughout the manuscript for other acronyms.

6. Line 188. What was the reason of choosing particular density?

Response: The rBC density with value of 1.8 g/cm3 utilized in this study is referred to relevant literature (Bond and Bergstrom, 2006). This density was also widely used in SP2-related studies (e.g., Schwarz et al., 2013; Gong et al., 2016). There are also many studies chose the value of 2 g/cm3 for the rBC density (e.g., Wang et al., 2016). We have considered the difference in rBC density when compared to the size distributions shown in other studies.

7. Line 195. The relatively similar...

Response: We have modified the statement there in the revision according to the comment.

8. Line 207. This is clearly the most likely reason for different VED values. Consider also cloud processing (~15min time scale) initiating wet deposition.

Response: Actually, we also consider the dry/wet deposition of large rBC particles result in the small VED values of rBC measured at the remote regions where local emissions are rare. In the manuscript, we just cited the initial interpretation for the small rBC presented in the corresponding reference in which they considered the small rBC was result from the source. Please note that we are referring to the rBC core inside an aerosol particle while the reviewer frequently referred to the ambient aerosol. The rBC cores can be coated by other components through aging processes in the atmosphere regardless of the initial size of rBC cores. Coating alters the physical/chemical of the initial rBC particle, including the geometric/optical size and the hydrophilicity of the particle in which the rBC core is embedded. Thus, the coated rBC particles should all undergo the cloud processing no matter the initial rBC cores are large or small.

9. Line 214. It is not much smaller, only smaller by 10%.

Response: The size distribution of rBC cores was presented in this study rather than the ambient aerosol particle. In this sense, 10% is significant. Aging processing changed the size of rBC-containing particles and had less impact on the mass equivalent size of rBC cores. The difference in peak diameter of SP2 derived rBC during polluted and clean days was significant, although the relative deviation was ∼10% (221 nm on polluted day vs. 199 nm on clean day).Wang et al. (2016) also showed a significant difference in rBC peak diameter on coal combustion (215 nm) and traffic (189 nm) dominant days. In the revised the manuscript, we have used 'obviously' instead of the initial 'much' to make the statement here more rigorous.

10. Line 230. Generally, VED(rBC) were positively correlating with AS/EC and AN/EC ratios....

Response: We have modified the statement there in the revision according to the comment.

11. Line 242. Availability of ammonia is most likely responsible for such patter, because sulphuric acid is neutralized first and only then nitric acid (acid strength effect) if there is enough ammonia. Neutralisation with ammonia is a passive process. Try looking at differences in the degree of neutralisation.

Response: Generally, there was enough ammonia to neutralize sulphuric acid and nitiric acid in most samples except in several severely polluted samples (Fig. 1). Even in the severely polluted samples, the amount of ammonia was enough to neutralize sulphuric acid. In this study, we simply assumed there was enough ammonia to neutralize both sulphuric acid and nitiric acid. The mass concentrations of ammonia sulfate were calculated from those of sulfate multiplying by a factor of 1.375, and those of ammonia nitrate were from nitrate multiplying by a factor of 1.29, according to the molecular weight. The AS/EC ratios should be constant even the neutralization was considered because there was enough ammonia to neutralize sulphuric acid first. Although the AN/EC ratios in severely samples might decrease because there was not enough residual ammonia to neutralize nitric acid entirely, it had little effect on the results and conclusion there (Figure 2b and 2d in the revision).

12. Line 252. How could secondary AN have any effect on the core size of rBC when

formation processes of the two are completely different?

Response: The statement in the initial manuscript was indeed confusing. Actually, we intended to express that the secondary formation of AN played an important role in the coating processes of rBC while had negligible effect on the core size of rBC. The positive correlation between VED values of rBC and AN/EC ratios was resulted from the simultaneous effect of source alteration. Regional transport brought larger rBC particles as well as higher AN/EC. Secondary formation itself should not affect the core sizes of rBC. We have revised in the revised manuscript to this: "The secondary formation of AN might play an important role in the coating processes of rBC but have a negligible effect on the core size of the rBC."

13. Line 274. Reference London study.

Response: We have added the reference in the revision.

14. Line 315. "which has little coating. . ."

Response: Revised.

15. Line 316. Why is this surprising as large VED(rBC) would be formed in the presence of copious amounts of gaseous precursors contributing to thick coating during atmospheric processing (secondary formation)?

Response: The VED of rBC presented in this study indicated the mass equivalent size of rBC cores. Thus, even coating can increase the size of an entire particle in which rBC core was embedded, the mass size of rBC core should not vary. If only the coating processing had effect, the VEDrBC should vary little while NFcoated gradually increase. The linear relationship between VEDrBC and NFcoated should relate to the rBC source alteration. We are surprised that almost the same linear relationship was observed in another winter campaign, having similar slope and y-intercept values. Thus, we suspected the linear relationship between VEDrBC and NFcoated might be a common result at our observation site.

16. Line 351. The method ignores the fact that local traffic VED is including contribution of other sources unless the authors have access to specific experiments proving the traffic VED. Considering contribution of other sources to VED makes local traffic contribution biased high. The method works for the estimating the upper limit, but the real contribution can be very different without proper measurement of traffic emitted VED (rBC).

Response: We quite agree with the reviewer's comments here. Actually, the VED of rBC from local traffic was inferred from the observed linear relationship between VEDrBC and NFcoated, by assuming that the VEDrBC of non/thickly coated (i.e., non or thinly coated) rBC particles was from local traffic. As the freshly emitted rBC was gradually coated through aging processes in the atmosphere, the completely non/thinly coated rBC was seldom observed at an ambient site. The observed minimum NFcoated with value of ∼10% at 5 minutes resolution (Figure 4) also indicated that there was no absolutely non/thinly coated rBC unless measured close to the traffic source (i.e., car tailpipe). This VED presented in this study is an inferred value and cannot be directly proved at this stage because of the limitation in the direct measurement of traffic VED. As mentioned in the response to the first major comment, size-segregated EC distributions revealed a typical traffic emitted EC mode with mass aerodynamic diameter of ∼150 nm in China, although another traffic emitted EC mode with diameter of ∼400 nm was also reported which was seldom observed in urban areas of developed countries (Kleeman et al., 2000; Allen et al., 2001; Huang et al., 2006, 2008; Yu and Yu, 2009; Yu et al., 2010). Further direct measurement to the VED of rBC from certain source is needed. However, the interpretation in our study is scientifically sound. Coal combustion, biomass burning and traffic exhaust should be the most major sources of the airborne rBC. Due to the combustion condition and fuel type, the rBC particles from coal combustion and biomass burning are most thickly coated by organic or inorganic components. In contrast, rBC particles freshly emitted from traffic source are usually naked and become thinly coated in a short time after emitted to the atmosphere. They should take several hours or longer to get thick coating. Thus, on this basis, we consid-

ered the VED of completely non/thinly coated rBC corresponded to traffic induced rBC. It was further confirmed by a very similar result observed in another independent campaign. As we illustrated in the conclusion section in our revised manuscript, "despite potential large uncertainties in the estimated contribution from the local traffic to rBC, due to the many assumptions employed, its relative variation is clearly demonstrated." Further research measuring sizes of rBC directly from various sources, including coal combustion, biomass burning and traffic exhaust, is needed to validate the findings presented in this study.

17. Line 357. This is a very bold assumption when BC sources are not localized, but instead present in every region. Differences in VED in different regions depend on relative contribution of sources, but never fixed. Also consider heterogeneity of traffic sources as car fleet in major cities is very different from rural ones.

Response: Indeed, the assumption here is very bold. However, it is very difficult to identify the rBC sources when they are not localized. Carbon isotope method is suitable to identify whether rBC from fossil fuel or biomass burning, however, it is also difficult to distinguish the traffic source from coal or other fossil oil combustion. A simple method based on SP2 measurement was thus proposed in this study, although with uncertainties. We considered that the contribution of different rBC sources (coal combustion/biomass burning) in air masses from a certain direction should vary little since the emission factors should be spatially and temporally stable during a short period (e.g., several days). We didn't focus on how much the coal combustion/biomass burning contributed. As there were no sources of local coal combustion/biomass burning in urban Beijing, we treated the coal combustion/biomass burning as an ensemble source from regional transport which was distinguishable from local traffic according the rBC sizes. The rBC sizes from such ensemble source were mainly determined by where these rBC particles were from. Thus, the assumption here is reasonable to some degree. We presented the analysis of local traffic contribution as a discussion section, indicating there were inevitable uncertainties during the analysis. We cannot give a

definite conclusion at current stage. In addition, considering the urban location of our observation site, the local traffic source referred in particular to the traffic emissions on the roads in urban Beijing in this study.

18. Line 370. Five typical VEDs were identified in VED range of 5% only? This is unreasonable.

Response: Considering the relative difference of rBC peak diameter on polluted and clean days is only 10%, the VED range of 5% is acceptable. The sizes of rBC core are generally small and relative stable. Due to their chemical inertia, the rBC sizes are mainly determined by sources and depositions. The five typical VEDs estimated in this study represent the rBC sizes from sources other than local traffic. The rBC sizes from these sources should be relatively stable and influenced by the contribution of coal combustion/biomass burning as well as the deposition processing. We cannot distinguish the contribution of coal combustion and biomass burning from these sources other than local traffic. During our twenty days campaign, the emission inventory should vary little and significant wet scavenging wasn't observed. Thus, the difference with relative value of 5% can be expected.

19. Line 435. Those values must correlate, because they are methodologically related.

Response: Although the size and mass of rBC are methodologically related, the negative correlation between them is not inevitably. If the increase in mass concentration of rBC only induced by the meteorological condition (e.g., decrease in mixing layer height), the relative contribution of traffic to rBC should varied little. Change a way of thinking, if we used the mass concentration of PM2.5 measured using a real-time PM2.5 monitor instead of rBC, the contribution of traffic should also negatively correlated with the mass concentration of PM2.5 because the mass concentrations of PM2.5 and rBC are well positively correlated (Wu et al., 2016).

20. Why there is a gap in the spectrum?

Response: The gap is related to the methodology. The SP2 used in this study is revision C*. It has eight channels. The incandescence signals are simultaneously captured at high and low gain channels. The high gain channel provides a much sensitive measurement of small rBC particles while the low gain extend the up limitation of SP2. There is a merging of dataset recorded by the two channels, resulting in abnormal values at the junction. The gap in the spectrum is resulted from the artificial elimination of those abnormal values before curve fitting.

21. Fig.5 (and respective text). Sources are poorly separated, because of high traffic contribution around midnight. That is unreasonable and points at overestimated traffic contribution.

Response: The high rBC concentration of traffic in night time is greatly induced by the depressing of mixing height and suppressing of turbulence. Besides, although the vehicle number on the road decreases in night time, heavy-duty diesel vehicles are permitted to travel in the urban area of Beijing during the period of 23:00 to 06:00 (local time) under Beijing's traffic regulation (Song et al., 2012). They have much higher emission factors than light-duty gasoline vehicles, also resulting in the high traffic BC concentration around midnight (Song et al., 2013).

References:

Allen, J. O., Mayo, P. R., Hughes, L. S., Salmon, L. G., and Cass, G. R.: Emissions of size-segregated aerosols from on-road vehicles in the Caldecott Tunnel, Environ. Sci. Technol., 35, 4189–4197, 2001. Bond, T. C., and Bergstrom, R. W.: Light absorption by carbonaceous particles: an investigative review, Aerosol Sci. Technol., 40, 27–67, 2006.

Bond, T. C., Doherty, S. J., Fahey, D. W., et al.: Bounding the role of black carbon in the climate system: A scientific assessment, J. Geophys. Res. Atmos., 118, 5380–5552, 2013. Cao, J. J., Chow, J. C., Tao, J.: Stable carbon isotopes in aerosols from Chinese cities: Influence of fossil fuels, Atmos. Environ., 45, 1359–1363, 2011.

Gong, X. D., Zhang, C., Chen, H., Nizkorodov, S. A., Chen, J. M., and Yang, X.: Size distribution and mixing state of black carbon particles during a heavy air pollution episode in Shanghai, Atmos. Chem. Phys., 16, 5399–5411, 2016.

Huang, X. F., Yu, J. Z., He, L. Y., and Hu, M.: Size distribution characteristics of elemental carbon emitted from Chinese vehicles: results of a tunnel study and atmospheric implications, Environ. Sci. Technol., 40, 5355–5360, 2006.

Huang, X. F., and Yu, J. Z.: Size distributions of elemental carbon in the atmosphere of a coastal urban area in South China: characteristics, evolution processes, and implications for the mixing state, Atmos. Chem. Phys., 8, 5843–5853, 2008.

Kleeman, M.J., Schauer, J. J., and Cass, G. R.: Size and composition distribution of fine particulate matter emitted from motor vehicles, Environ. Sci. Technol. 34(7), 1132–1142, 2000.

Laborde, M., Mertes, P., Zieger, P., Dommen, J., Baltensperger, U., and Gysel, M.: Sensitivity of the single particle soot photometer to different black carbon types, Atmos. Meas. Tech., 5, 1031–1043, 2012.

Liu, J. W., Mo, Y. Z., Li, J.: Radiocarbon-derived source apportionment of fine carbonaceous aerosols before, during, and after the 2014 Asia-Pacific Economic Cooperation (APEC) summit in Beijing, China, J. Geophys. Res. Atmos., 121, 4177–4187, 2016.

Moteki, N., and Kondo, Y.: Effects of mixing state on black carbon measurements by laser-induced incandescence, Aerosol Sci. Technol., 41, 398–417, 2007.

Schwarz, J. P., Gao, R. S., Perring, A. E., Spackman, J. R., and Fahey, D. W.: Black carbon aerosol size in snow, Sci. Rep., 3, 1356, doi:10.1038/srep01356, 2013.

Song, S., Wu, Y., Jiang, J., et al.: Chemical characteristics of size-resolved PM2.5 at a roadside environment in Beijing, China, Environ. Pollut., 161, 215–221, 2012.

Song, S., Wu, Y., Xu, J., et al.: Black carbon at a roadside site in Beijing: Temporal

variations and relationships with carbon monoxide and particle number size distribution, Atmos. Environ., 77, 213–221, 2013.

Wang, Q. Y., Huang, R. J., Zhao, Z. Z., Cao, J. J., Ni, H. Y., Tie, X. X., Zhao, S. Y., Su, X. L., Han, Y. M., Shen, Z. X., Wang, Y. C., Zhang, N. N., Zhou, Y. Q., and Corbin, J. C.: Physicochemical characteristics of black carbon aerosol and its radiative impact in a polluted urban area of China, J. Geophys. Res. Atmos., 121, doi:10.1002/2016JD024748, 2016.

Widory, D.: Combustibles, fuels and their combustion products: A view through carbon isotopes, Combust. Theor. Model., 10, 831–841, 2006.

Wu, Y. F., Zhang, R. J., Tian, P., et al..: Effect of ambient humidity on the light absorption amplification of black carbon in Beijing during January 2013, Atmos. Environ., 124, 217–223, 2016.

Yu, H., and Yu, J. Z.: Modal characteristics of elemental and organic carbon in an urban location in Guangzhou, China, Aerosol Sci. Technol., 43, 1108–1118, 2009.

Yu, H., Wu, C., Wu, D., and Yu, J. Z.: Size distributions of elemental carbon and its contribution to light extinction in urban and rural locations in the Pearl River Delta region, China, Atmos. Chem. Phys., 10, 5107–5119, 2010.

Zhang, Y. L., Huang, R. J., El Haddad, I., et al.: Fossil vs. non-fossil sources of fine carbonaceous aerosols in four Chinese cities during the extreme winter haze episode of 2013, Atmos. Chem. Phys., 15, 1299–1312, 2015.

Please also note the supplement to this comment:
http://www.atmos-chem-phys-discuss.net/acp-2016-1096/acp-2016-1096-AC1-supplement.pdf
* * *
[Figure]

**Fig. 1.** Correlations between cations and anions: (a) NH4+ versus SO42-+NO3-; (b) Cations versus Anions

---

## Author Comment (AC2) · 27 Apr 2017

We appreciate the criticisms from the reviewer, although we cannot agree completely as explained below. We also appreciate the several useful comments which helped us improve the paper.

1. The reviewer felt that the scientific novelty value of our results was poor and mentioned that there was a similar dataset published by Wang et al. (2016). Moreover, the reviewer pointed out that there had been already several other articles containing SP2 measurement results published from Beijing and urban areas in Asia with longer data time series and much larger set of instruments, and results from these relevant studies have include the main results of our work. A longer dataset with data collected during different seasons and a larger set of instruments was recommended.

[Figure]

Response:

We are fully aware of the article by Wang et al. (2016a) in the preparation of our manuscript. Wang et al. (2016a) presented the SP2-rBC measurement that was collected right before our campaign in Beijing. The study only revealed the mass concentration of rBC derived from the SP2 measurement, and focused on the contribution of regional transport to the BC mass concentration in Beijing on the basis of WRF-BC model analysis combined with the SP2 measurement. The size distribution and mixing state of rBC particles, which are important knowledge gaps on rBC related sciences, were discussed in our study but not mentioned in their study. Thus, our study and their study have completely different focuses, although both focused on the same city. We'd also like to mention that there are two common coauthors in both papers whose contribution to our work was to provide guidance on instrument operation and data analysis. Our study is independent of that presented by Wang et al. (2016a) and we present more specific and advanced information of rBC that SP2 can provide, including size distribution and mixing state. We have reviewed many published SP2-related studies made in the urban areas in Asia, especially those in China. Studies on the size distribution of rBC, are relatively limited in China, e.g., Pearl River Delta Region (Huang et al., 2011, 2012), Yangtze River Delta (Gong et al., 2016), Xi'an in West China (Wang et al., 2015), Qinghai-Tibetan Plateau (Wang et al., 2014), and no similar study using SP2 was available in urban Beijing, or the whole North China Plain (NCP) where Beijing is located. Since rBC size distribution varied substantially from region to region, as reviewed in Huang et al. (2012), conducting such a study as presented in our manuscript is needed and filled the knowledge gap on this important scientific topic. As for the length of the campaign the reviewer concerned, we'd like to mention that measuring the rBC for a lengthy period with SP2 is challenging mainly due to its high cost, as well as complexity in operation and maintenance. Most in-situ SP2 measurements were performed in a period shorter than one-month (e.g., Liu et al., 2014; Huang et al., 2011; Wang et al., 2014; Gong et al., 2016; . . .). Sometimes a number of instruments were operated accompanied with SP2 to show richer aerosol

characteristics. One of our previous work used a photoacoustic extinctiometer (PAX, DMT, USA) operating parallel to the SP2 to investigate the effect of mixing state on the mass absorption efficiency of rBC (Wu et al., 2016). The related in-situ experiment of the previous work was made at the same site in January 2013 when the heavy haze events occurred frequently in Beijing. However, our current research aimed to reveal the rBC size characteristic of rBC in urban Beijing. The following discussion on the sources of rBC in this study was mainly based on the measured rBC sizes. Actually, the size distributions of rBC is very similar in our two campaigns (the previous one was in the winter of 2013 and the current one was in the winter of 2014), because both measurements were made at the same site. They have been presented in Fig. S2 in the revised supplemental files. The slightly larger rBC sizes in the winter of 2013 is likely to relate with the higher rBC concentrations, which indicate more influence of larger rBC from sources other than local traffic, e.g., regional transport. In order to reveal the size distribution of rBC in urban Beijing and the causes of its variation, we focus on the campaign in the winter of 2014 in our current manuscript, during which the PM2.5 samples were collected twice per day and chemical compositions of each sample (including water-soluble ions, metal elements, carbonaceous matters) were comprehensively analyzed. In the discussion section of our current manuscript (Fig.4), we had already employed the dataset acquired in the winter of 2013 to verify the relationship between the number fraction of thickly coated rBC (NFcoated) and the average volume equivalent diameter of rBC (VEDrBC). Since the severe air pollution in Beijing is mainly occurred in wintertime, the rBC characteristics based on SP2 measurement in other seasons were seldom paid attention in our previous work. We will certainly consider the reviewer's suggestion in our future studies.

2. The reviewers mentioned that the develop method for analyzing the contribution of local traffic is poorly described and verified. Meanwhile, it is hard to estimate if it works. Proving the developed method works by comparing results of our method to other independent measurement data is recommended. The reviewer also suggests that it is necessary to estimate uncertainty and validate the method. Moreover, the

reviewer is also concerned with universality of this developed method.

Response:

We have described the method in more detail in the revised manuscript, according to the review's comments (Lines 398–400 in the revision). Generally, the method employed in our study is based on the extrapolative size of local traffic source from the linear relationship between the number fraction of thickly coated rBC (NFcoated) and the average volume equivalent diameter of rBC (VEDrBC). This diameter cannot be verified at current stage, because the directly measurement to rBC size of certain rBC source, e.g., traffic exhaust, is still lacking. However, as mentioned in section 4.1, through the analysis of the data acquired at the same site in the winter of 2013, we found that this relationship between NFcoated and VEDrBC and the inferred traffic-rBC diameter could be repeated. Moreover, several references associated with size-segregated aerosol samples were added in the revised manuscript (Lines 334–345 in the revision), because a similar BC size with diameter of ∼150 nm was also presented in these studies (Yu and Yu, 2009; Yu et al., 2010). Thus, we considered that this diameter actually existed in urban Beijing. Of cause, further work should be done to verify this result, especially the measurement nearby the emission sources and in other seasons. Actually, it is very difficult to distinguish the source of ambient rBC, as the rBC was easily to be internally mixed with other components during a short period in the ambient (Peng et al., 2016). Considering the chemical inertness of BC and invariance of the mass-equivalent size of the individual rBC particle in ambient atmosphere, we developed this simple method to estimate the contribution of local traffic to rBC. A more accurate source apportionment of rBC was presented by Liu et al. (2014) also using the difference in the size distribution of traffic-related rBC and solid fuel-related rBC. Their method is much more complicated. Based on the combination of PMF analysis, Wang et al. (2016b) developed a method to distinguish the rBC from traffic, coal combustion and biomass burning sources. A large amount of trace elements should be analyzed if the method was employed. Moreover, large uncertainties should also

be related in their method. The method developed in our study is relatively simple and requires only the SP2 measurement itself. We admit that large uncertainties existed in the resolved contribution of different sources to BC on the basis of our method, as explained in the revised manuscript. Several other methods, such as radiocarbon analysis, are generally used to determine the source of BC. However, the 14C analysis is mainly used to distinguish the contribution of biomass burning and fossil-fuel combustion to BC (Zhang et al., 2015; Liu et al., 2016). It is difficult to distinguish the traffic-related source from the fossil-fuel combustion based on the 14C analysis (see the response to the second major comment in RC1 for detail). Actually, the method developed in our study to resolve the rBC sources is more qualitative than quantitative, as several rough assumptions were employed in this method. However, results from this method should well reflect the variation in the traffic contribution. According to the results resolved from this method, it was much clearer that a significant increase in the contribution of sources other than local traffic, e.g., regional transport, was observed during the haze period. Based on many assumptions, uncertainty in the resolved traffic contribution should be very large and cannon be simply quantized here. As the sources of rBC are discrepant in different regions, we are not very sure this method is suitable in other studies. However, considering the rBC freshly emitted from the local traffic source is mostly non/thinly coated, the extrapolation of local traffic-rBC size by assuming it equal to the VEDrBC of totally non/thinly coated rBC should be tried in the typical urban regions at least. The rBC source apportionment method developed in our study can also be tried in the local traffic dominated urban areas where other rBC sources, e.g., coal combustion and biomass burning, are rare.

3. Why the diurnal cycles of the BC from both traffic and other sources are highest during nighttime?

Response:

As mention in section 4.2 in the initial manuscript, the higher BC concentrations during nighttime were likely to be attributed to the lower mixing layer height. Obvious decrease

in the mixing layer height suppressed the diffusion of air pollutants, resulting the higher BC from both traffic and other sources during nighttime. Another important cause of the higher traffic-BC during nighttime might be the increase in the flow of heavy-duty diesel vehicles. These vehicles have much higher emission factors of BC and are permitted to travel in urban Beijing only during the night time from 23:00 to 06:00 Local Time (Song et al., 2013). We have added such information in the revised manuscript with relevant references (Lines 432–436 in the revision).

4. Uncertainty estimation for BC results and especially for the larger mode would be important addition. The large BC mode (with maximum at around 500nm) is very small and very close to the upper particle size limit of the instrument, so it would be important to estimate how real this BC mode is.

Response:

Thanks for this recommendation. As mentioned in the methodology section, the overall uncertainty in the rBC mass determination was ∼25%, including the uncertainties inherented in the mass calibration, flow measurement and estimation of BC masses beyond the SP2 detection range. Indeed, the large BC mode is very small and very close to the upper size limit of the SP2. At the initial stage of analyzing this dataset, we also suspected the reliability of the large rBC mode. We then first consulted a large amount of relevant references and found that the large mode was also observed in several previous studies as mentioned in the first paragraph of section 3.1 in our manuscript (Huang et al., 2011; Wang et al., 2014). We still make reservations on whether the large rBC mode (only accounting for ∼6% of the SP2-determined rBC masses) was related to the inherent measurement bias, as it is close to the upper size limit of the SP2. Thus, we chose not to focus on the large mode in our manuscript discussion, and leave this to future more accurate measurements. Since there is no explicit interpretation to this large mode in literature, we simply present the observational facts from the dataset without much discussions.

5. The title and aim of this article are slightly conflicting. Title suggest that this article focuses on BC size distributions and sources, whereas the aim states that the aim was to develop a novel approach to evaluate the contribution of local traffic to the rBC concentration. Might be useful to include the method development to the title, if that is the main goal.

Response:

Actually, the major goal of this study is to reveal the size distribution of rBC and analyze its variation in urban Beijing based on the SP2 measurement. The source of rBC is a second goal of the study. We have modified the last paragraph of the introduction section to make the subject of this study clearer according to the suggestion (Lines 125–133 in the revision).

References:

Gong, X. D., Zhang, C., Chen, H., Nizkorodov, S. A., Chen, J. M., and Yang, X.: Size distribution and mixing state of black carbon particles during a heavy air pollution episode in Shanghai, Atmos. Chem. Phys., 16, 5399–5411, 2016.

Huang, X. F., Gao, R. S., Schwarz, J. P., He, L. Y., Fahey, D. W., Watts, L. A., Mc-Comiskey, A., Cooper, O. R., Sun, T. L., Zeng, L. W., Hu, M., and Zhang, Y. H.: Black carbon measurements in the Pearl River Delta region of China, J. Geophys. Res., 116, D12208, doi:10.1029/2010JD014933, 2011.

Huang, X. F., Sun, T. L., Zeng, L. W., Yu, G. H., and Luan, S. J.: Black carbon aerosol characterization in a coastal city in South China using a single particle soot photometer, Atmos. Environ., 51, 21–28, 2012. Liu, D., Allan, J. D., Young, D. E., Coe, H., Beddows, D., Fleming, Z. L., Flynn, M. J., Gallagher, M. W., Harrison, R. M., Lee, J., Prevot, A. S. H., Taylor, J. W., Yin, J., Williams, P. I., and Zotter, P.: Size distribution, mixing state and source apportionment of black carbon aerosol in London during wintertime, Atmos. Chem. Phys., 14, 10061–10084, 2014.

[Figure]

Liu, J. W., Mo, Y. Z., Li, J.: Radiocarbon-derived source apportionment of fine carbonaceous aerosols before, during, and after the 2014 Asia-Pacific Economic Cooperation (APEC) summit in Beijing, China, J. Geophys. Res. Atmos., 121, 4177–4187, 2016.

Peng, J. F., Hu, M., Guo, S., Du, Z. F., Zheng, J., Shang, D. J., Zamora, M. L., Zeng, L. M., Shao, M., Wu, Y.-S., Zheng, J., Wang, Y., Glen, C. R., Collins, D. R., Molina, M. J., and Zhang, R. Y.: Markedly enhanced absorption and direct radiative forcing of black carbon under polluted urban environments, Proc. Natl. Acad. Sci. U.S.A., 113, 4266–4271, 2016.

Song, S., Wu, Y., Xu, J., et al.: Black carbon at a roadside site in Beijing: Temporal variations and relationships with carbon monoxide and particle number size distribution, Atmos. Environ., 77, 213–221, 2013.

Wang, Q. Y., Schwarz, J. P., Cao, J. J., Gao, R. S., Fahey, D. W., Hu, T. F., Huang, R. J., Han, Y. M., and Shen, Z. X.: Black carbon aerosol characterization in a remote area of Qinghai–Tibetan Plateau, western China, Sci. Total Environ., 479–480, 151–158, 2014.

Wang, Q. Y., Liu, S. X., Zhou, Y. Q., Cao, J. J., Han, Y. M., Ni, H. Y., Zhang, N. N., and Huang, R. J.: Characteristics of Black Carbon Aerosol during the Chinese Lunar Year and Weekdays in Xi'an, China, Atmosphere, 6, 195–208, 2015.

Wang, Q. Y., Huang, R. J., Cao, J. J., et al.: Contribution of regional transport to the black carbon aerosol during winter haze period in Beijing, Atmos. Environ., 132, 11–18, 2016a.

Wang, Q. Y., Huang, R. J., Zhao, Z. Z., Cao, J. J., Ni, H. Y., Tie, X. X., Zhao, S. Y., Su, X. L., Han, Y. M., Shen, Z. X., Wang, Y. C., Zhang, N. N., Zhou, Y. Q., and Corbin, J. C.: Physicochemical characteristics of black carbon aerosol and its radiative impact in a polluted urban area of China, J. Geophys. Res. Atmos., 121, doi:10.1002/2016JD024748, 2016b.

Wu, Y. F., Zhang, R. J., Tian, P., et al..: Effect of ambient humidity on the light absorption amplification of black carbon in Beijing during January 2013, Atmos. Environ., 124, 217–223, 2016.

Yu, H., and Yu, J. Z.: Modal characteristics of elemental and organic carbon in an urban location in Guangzhou, China, Aerosol Sci. Technol., 43, 1108–1118, 2009.

Yu, H., Wu, C., Wu, D., and Yu, J. Z.: Size distributions of elemental carbon and its contribution to light extinction in urban and rural locations in the Pearl River Delta region, China, Atmos. Chem. Phys., 10, 5107–5119, 2010.

Zhang, Y. L., Huang, R. J., El Haddad, I., et al.: Fossil vs. non-fossil sources of fine carbonaceous aerosols in four Chinese cities during the extreme winter haze episode of 2013, Atmos. Chem. Phys., 15, 1299–1312, 2015.

Please also note the supplement to this comment:
http://www.atmos-chem-phys-discuss.net/acp-2016-1096/acp-2016-1096-AC2-supplement.pdf

---

## Author Response (AR2)

The authors made a genuine effort in responding to the reviewers concerns and overall most of the issues have been adequately addressed. However, I am surprised how little of the response material has been included into the revised manuscript. Reviewers typically raise questions/comments not for themselves, but rather from the readers perspective. I recommend that a lot more of the response material would be included into the revised manuscript. For example, the response on the neutralization degree can briefly be mentioned in the text and the Figure included into Supplementary Material; discussion on validation of the proposed method by carbon isotope analysis did not make into revised manuscript; and so on.

Response:

We again appreciate the reviewer's further comments. We have added more discussions in the final version of the paper based on this recommendation, such as including the discussion on the neutralization degree, and the method of calculating mass concentrations of ammonia sulfate and ammonia nitrate (Lines 239–241). The corresponding Figure has also been added in the supplemental document (Fig. S3).

The carbon isotope analysis is a good method to identify the source of BC; however, as mentioned in our previous response to the referees, this method was usually used to distinguish the fossil-fuel combustion and biomass burning produced BC. It is difficult to distinguish the traffic-related BC from other fossil-fuel combustion sources, e.g., coal combustion. Thus, we only briefly compared our result to that based on carbon isotope analysis in the revised manuscript (Lines 409–411).

Response on defining thickly coated particles makes sense now and I suggest to look at the rBC size of the thinly coated particles. This presents some sort of validation (the major deficiency of the paper) where traffic related rBC core size derived from regression should roughly match the core size of thinly coated particles. Considering urban Beijing area having no major industrial sources, traffic related particles should stay thinly or non coated due to a short residence time after being emitted. Two peaks in lag time suggest two distinct populations with the traffic related particles clustering around short lag time while traffic related particles from long ranges and produced by biomass burning possessing larger lags (the more severe pollution the larger the percentage of thickly coated particles). Therefore, while the regression may indeed point correctly at traffic related rBC cores having VED of 150nm, similar VED should be revealed by thinly coated particles which should arise from nearby traffic sources. My question is what is the rBC VED of the particles having a lag of approximately 1us and 0us? If the above question is positively responded it would significantly strengthen the paper.

Response:

We greatly appreciate the reviewer for providing this constructive comment, which provided us an idea for strengthening the paper. Accordingly, we calculated the VED of non/thinly coated rBC (lag time shorter than 2 μs) with 5-min resolution and presented its frequency histogram in Fig. S4 as a supplemental material (also shown below). The frequency histogram can be well fitted by a Gauss function with the peak diameter of 149.2 nm. The value is almost the same as the mean of the entire 5-min VEDs of non/thinly coated rBC (149.5 nm). Both the peak diameter of gauss fitting and the mean diameter are close to the deductive VED of rBC from local traffic source (~150 nm) discussed in our manuscript.  Such a finding further proves our hypothesis and conclusions already discussed in the paper, since the non/thinly coated rBC particles in urban Beijing are likely most from the local traffic exhausts. We have added a brief discussion based on this finding in the final version of the paper (Lines 329–334).

[Figure]

**Fig. S4.** Frequency histogram of the 5-min average volume-equivalent diameters (*VED*) of rBC without or with thin coating (black line). Gauss fitting is performed to the histogram (red line). Boxplot of the *VED* of non/thinly coated rBC is also overlapped (blue box).

**Marked-up manuscript version**

[revised manuscript text omitted]